# Structural basis for recognition of transcriptional terminator structures by ProQ/FinO domain RNA chaperones

Hyeong Jin Kim [1], Mazzen Black [1], Ross A. Edwards[1], Flora Peillard-Fiorente [2], Rashmi Panigrahi [1], David Klingler [3], Reiner Eidelpes [3], Ricarda Zeindl [3], Shiyun Peng[1], Jikun Su[1], Ayat R. Omar[1], Andrew M. MacMillan[1], Christoph Kreutz [3], Martin Tollinger [3], Xavier Charpentier [2], Laetitia Attaiech [2] ✉ & J. N. Mark Glover [1] ✉

The ProQ/FinO family of RNA binding proteins mediate sRNA-directed gene regulation throughout gram-negative bacteria. Here, we investigate the structural basis for RNA recognition by ProQ/FinO proteins, through the crystal structure of the ProQ/FinO domain of the *Legionella pneumophila* DNA uptake regulator, RocC, bound to the transcriptional terminator of its primary partner, the sRNA RocR. The structure reveals specific recognition of the 3' nucleotide of the terminator by a conserved pocket involving a β-turn-α-helix motif, while the hairpin portion of the terminator is recognized by a conserved α-helical N-cap motif. Structure-guided mutagenesis reveals key RNA contact residues that are critical for RocC/RocR to repress the uptake of environmental DNA in *L. pneumophila*. Structural analysis and RNA binding studies reveal that other ProQ/FinO domains also recognize related transcriptional terminators with different specificities for the length of the 3' ssRNA tail.

Small RNAs (sRNAs) control a variety of physiological responses across bacterial species[1]. sRNAs usually work by pairing with target mRNAs, often with the assistance of protein partners called RNA chaperones. RNA chaperones, such as the well-studied Hfq[2–4], play roles in either increasing rates of sRNA–mRNA hybridization, affecting the stability of target mRNA, or rearranging target RNA folding impacting ribosome accessibility[4].

The proteins containing a ProQ/FinO domain [https://www.ebi.ac.uk/interpro/entry/pfam/PF04352/] constitute a large family of RNA chaperones that are widely distributed throughout the bacterial taxa[5–8]. This family is defined by a conserved ProQ/FinO domain, which is a largely α-helical fold that is often flanked by flexible N- or C-terminal regions[6,9]. The eponymous FinO protein was discovered as a regulator of F plasmid conjugation nearly 50 years ago, and acts to bind a single partner sRNA called FinP to stabilize FinP and facilitate its interactions with its antisense partner, the mRNA encoding the major F plasmid transcription factor, TraJ[5]. More recently, the application of RNA-seq technologies has enabled the elucidation of the biological partners and targets of several ProQ/FinO family proteins. These approaches have verified the recognition of FinP by FinO, but have also revealed that FinO, in certain instances, can also interact with related sRNAs from other co-resident plasmids[10]. Another plasmid-encoded ProQ/FinO family member, FopA, has also been shown to interact with a single antisense RNA[11]. Likewise, the ProQ/FinO domain-containing protein RocC of *Legionella pneumophila* interacts with only one *trans*-acting sRNA (RocR) to repress post-transcriptionally multiple mRNA targets[6]. While these studies have built the case that certain ProQ/FinO family members bind to only a very limited number of partners in a

[1]Department of Biochemistry, University of Alberta, Edmonton, AB T6G2H7, Canada. [2]CIRI, Centre International de Recherche en Infectiologie, Team "Horizontal gene transfer in bacterial pathogens", Inserm, U1111, Université Claude Bernard Lyon 1, CNRS, UMR5308, Ecole Normale Supérieure de Lyon, Université de Lyon, 69100 Villeurbanne, France. [3]Institute of Organic Chemistry, Center for Molecular Biosciences Innsbruck (CMBI), University of Innsbruck, Innrain 80/82, 6020 Innsbruck, Austria. ✉e-mail: laetitia.attaiech@univ-lyon1.fr; mark.glover@ualberta.ca

highly specific manner, other family members appear to have multiple biological sRNA targets. Probably best studied is ProQ, which binds hundreds of sRNAs and likely acts as a general regulator of gene expression with an impact similar to Hfq[12–14]. Similarly, a minimal ProQ/FinO domain protein, NMB1681, has been shown to bind a range of structured RNAs in *Neisseria meningitidis*[15].

Insight into how these proteins recognize their cognate RNAs initiated with FinO. Early studies showed that FinO specifically binds the 3′ transcriptional terminator structure of FinP in a manner that critically relies on both the GC-rich hairpin and 3′ polypyrimidine tail that define the terminator[16], and further work showed that the base of the hairpin and 3′ tail are both strongly protected from ribonuclease degradation by FinO[17]. Proteolytic mapping and protein deletion analysis revealed that the ProQ/FinO domain itself is responsible for transcriptional terminator binding[18,19], and site-specific protein–RNA cross-linking identified key RNA contact surfaces on the core ProQ/FinO domain[20]. Further studies showed that the ProQ/FinO domains of ProQ and NMB1681, can also specifically recognize transcriptional termination structures[21,22].

More recent studies probing the mechanism of binding of ProQ/FinO proteins to biologically validated RNA targets have opened the possibility for more complex modes of RNA recognition. To date, ProQ has been most studied in this regard. RNA footprinting has suggested that while transcriptional terminators do appear to be protected, other regions far from the 3′ ends of the transcript can also be protected[23]. Interestingly, ProQ proteins also share another folded domain C-terminal to the ProQ/FinO domain that adopts a Tudor fold, and hydrogen-deuterium exchange experiments suggest that this region may also play a role in RNA binding[24]. Detailed recent work however confirms the key role of the ProQ/FinO domain of ProQ in the recognition of transcription terminators[25]. An analysis of the effects of ProQ point mutants on RNA binding in cells using a bacterial 3-hybrid approach indicated that a conserved concave surface on the ProQ/FinO domain could be the primary terminator binding surface[26]. Indeed, the same surface on the ProQ/FinO domain of FinO was found to interact with the FinP terminator, suggesting a common mechanism of interaction in the family[20].

While molecular modeling integrating small-angle X-ray scattering with biochemical and biophysical data has yielded low-resolution models for interactions of FinO[17] and ProQ[24] with RNA, the molecular mechanisms underlying these interactions remain unclear. To obtain high-resolution structural information, we chose to study one of the most selective of the ProQ/FinO domain protein family, the *L. pneumophila* protein RocC and its primary sRNA partner, RocR. The RocC/RocR system regulates competence, a specialized physiological state involving the coordinated expression of multiple genes which allows the bacteria to uptake environmental DNA and integrate it into its chromosome (i.e., natural transformation)[6]. RocC binds and stabilizes RocR, which in turn binds and represses a series of mRNA targets that encode components of the DNA uptake system. Previous work showed that the ProQ/FinO domain of RocC specifically recognizes the transcriptional terminator (Stem Loop 3, SL3) of RocR and that mutation of the ProQ/FinO domain abrogates RocC-mediated repression[6].

Here, we have determined the crystal structure of the ProQ/FinO domain of RocC bound to a minimal terminator derived from RocR. The structure reveals that the 3′ single-stranded RNA tail adopts a hook-like structure that docks the 3′ terminal nucleotide into a highly conserved pocket in the ProQ/FinO domain, while the hairpin portion of the RNA is recognized by a conserved α-helical N-cap structure that recognizes multiple contiguous phosphate groups on a single strand. The importance of the interactions visualized in the structure is supported by structure-guided site-directed mutagenesis with RNA binding and in vivo measures of competence. The structure also explains previous results from site-directed mutation studies on FinO and ProQ, suggesting a common mechanism of binding across the ProQ/FinO

domain family. We further present evidence that different ProQ/FinO domain proteins selectively bind terminators with different lengths of single-stranded tails.

## Results

### The ProQ/FinO domain of RocC specifically binds the SL3 stem-loop and 3′ tail of RocR

Previous work revealed that the N-terminal ProQ/FinO domain-containing region of RocC (RocC$_{1-126}$) could bind SL3 of RocR (noted hereafter RocR$_{SL3}$, Fig. 1a) with similar affinity to full-length RocC[6]. We carried out further truncations to define a minimal RNA-binding region of RocC that would be amenable to structural studies. Limited proteolysis suggested RocC$_{24-126}$ is a stable folded core, however, this construct was defective in interacting with RocR$_{SL3}$ and could not replace RocC in an in vivo assay to measure DNA uptake efficiency (Fig. 1b, c). In contrast, RocC$_{14-126}$ displayed a similar binding affinity to RocC$_{1-126}$ (Fig. 1b) and is perfectly functional in vivo (Fig. 1c), indicating that residues 14–23 contain critical residues for its interaction with RocR. We thus used RocC$_{14-126}$ for further EMSA studies.

Previous work on the FinO/FinP system indicated that FinO recognizes the base of the terminator hairpin and the 3′ ssRNA tail[16,17]. More recent studies have shown that the ProQ/FinO domain of *E. coli* ProQ binds similar hairpin-tail structures with at least a two base-pair stem and a 4-nucleotide 3′ tail consistent with the notion that ProQ, like FinO, binds mainly the base of the stem and 3′ tail[25]. To test the relative importance of the loop, stem, and 3′ tail of RocR$_{SL3}$, we generated various mutants and assessed their ability to bind RocC$_{14-126}$ by EMSA (Fig. 1 and Supplementary Figs. 1 and 2). Changing the size of the loop or reducing the stem length to five base pairs showed little effect on binding affinity, similar to what was observed for FinO/FinP (Supplementary Fig. 2a, b, d, e)[16,17]. However, reduction or elongation of the tail length of RocR$_{SL3}$ dramatically impacted binding affinity so that even a reduction or increase of the tail length by just one nucleotide reduced binding to the point that we could not see binding saturation under our conditions (Fig. 1b and Supplementary Fig. 2c, f). We also tested the possibility that RocC might recognize the specific sequence at the base of the hairpin and in the polypyrimidine tail (Fig. 1d and Supplementary Fig. 2c, g). Substitution of the GC-rich sequence at the base of the stem with AU base pairs did not significantly impact the affinity of binding, similar to previous results showing that FinO does not exhibit specificity for the base pairs at the base of the SL2 stem of FinP[16]. Likewise, the introduction of an adenine at either the 4th position or the terminal 5th position of the polypyrimidine tail did not significantly impact binding, suggesting a lack of sequence specificity in recognition of this region (Fig. 1d and Supplementary Fig. 2c, g).

### Overall structure of RocC bound to a terminator RNA

We established a method to obtain a homogenous protein–RNA complex of various RocC/RocR mutant combinations using gel filtration chromatography, and SEC-MALS confirmed the formation of a one-to-one complex (Supplementary Fig. 3). We crystallized and determined the structure of RocC both alone (Supplementary Fig. 4) and in complex with RocR$_{SL3}$ containing a modified stem-loop with a nine base-pair stem and a tetraloop (RocR$_{9bp-tet}$) at 3.2 Å (Fig. 2a). The asymmetric unit contains six copies of apo-RocC$_{14-126}$ and four copies of the protein–RNA complex. Overall, the hairpin-tail RNA binds to one side of the ProQ/FinO domain, largely through interactions between the protein and the RNA backbone. NCS averaged maps revealed electron density of sufficient quality to model the entire hairpin-tail RNA (Supplementary Fig. 5). As predicted (Supplementary Fig. 1d), the nine base-pair stem is fully paired, however, the 5′ single nucleotide, U$_1$, forms an additional mismatch base pair with 3′ U$_{24}$ and this base pairing (with the exception of the U–U pair) is confirmed by analysis of ¹H NMR spectra of the free and RocC-bound RocR$_{9bp-tet}$ (Supplementary Fig. 6). The hairpin adopts the expected A-form helical geometry

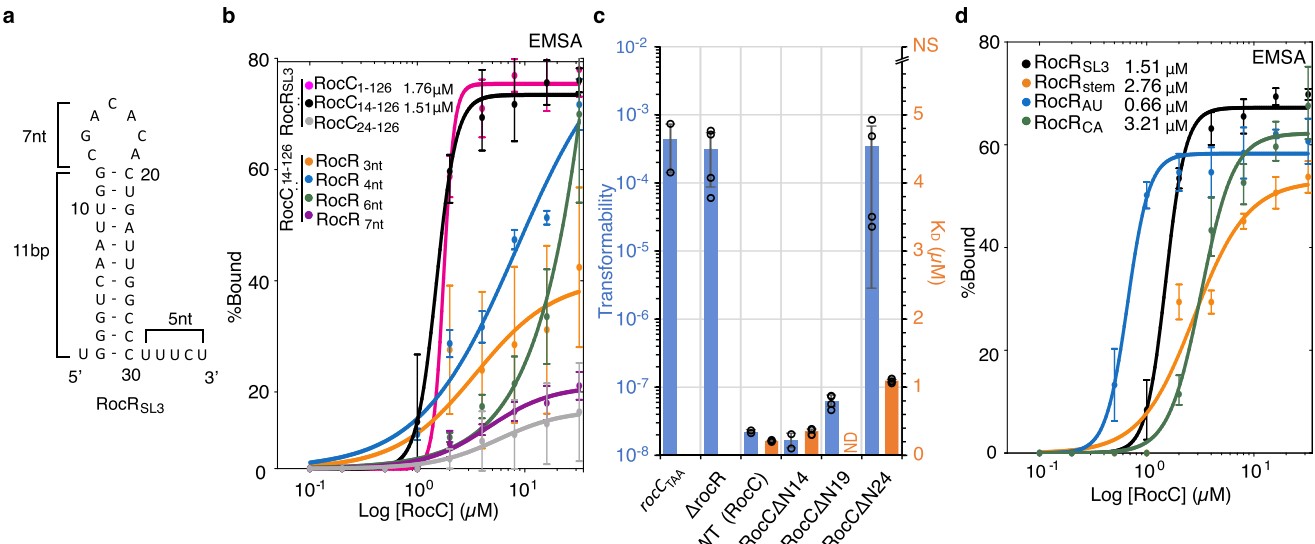

**Fig. 1 | Defining the determinants of RocC/RocR interaction in vitro and in vivo.**
**a** Predicted folding of wild-type RocR$_{SL3}$ using RNAfold web server. **b** EMSA binding assay for RocC$_{1-126}$, RocC$_{14-126}$, and RocC$_{24-126}$ vs 5′ radiolabeled RocR$_{SL3}$ ($n = 3$). Binding affinity of RocR$_{SL3}$ with different tail lengths was tested with RocC$_{14-126}$. The error bars are standard error of the mean (SEM). **c** Role of the N-terminus of RocC for RocR$_{SL3}$ binding in vitro and for uptake of DNA (transformation) in vivo. The transformation was assessed in *Legionella* strains containing either WT RocC, a mutant in which *rocC* translation is disrupted (*rocC$_{TAA}$*), a RocR deletion (Δ*rocR*), or different deletions at the N-terminus (RocCΔN14−deletion of a.a. 1–13; RocCΔN19 – deletion of a.a. 1-18; RocCΔN24−deletion of a.a. 1–23). Binding measurements were carried out with a fluorescence polarization (FP) assay using RocC$_{1–126}$ or the indicated N-terminal deletion mutants, with FAM-labeled RocR$_{SL3}$ as a target. The orange histograms indicate the K$_D$ values measured by FP ($n = 3$) and the blue histograms indicate the transformability of the indicated mutant or strain. Transformability is the ratio of the number of CFUs counted on selective medium divided by the number of CFUs counted on a non-selective medium. NS indicates mutants where RNA binding could not be detected. # indicates a mutant that was not tested in vitro. The error bars are the standard error of the mean (SEM). Transformation experiments were repeated at least twice on two independent clones of each mutant and with two types of donor DNA, FP experiments were repeated three times. **d** EMSA binding assay for RocC$_{14–126}$ with 5′ radiolabeled RocR$_{SL3}$ with various substitution mutants. The error bars are the standard error of the mean (SEM) ($n = 3$). Source data are provided as a Source Data file.

and the 5′-U$_{11}$U$_{12}$C$_{13}$G$_{14}$−3′ tetraloop[27], adopts the expected structure. The 4-nucleotide single-stranded 3′ tail (5′-U$_{25}$U$_{26}$C$_{27}$U$_{28}$−3′) adopts a hook-like structure. U$_{25}$ maintains an A-form geometry, however U$_{26}$ and C$_{27}$ bend away, unstacking from U$_{25}$ and instead stacking upon each other.

## Recognition of the RNA hairpin of RocR by an α helical N-cap motif in the ProQ/FinO domain of RocC

The hairpin portion of the RNA is bound by the N-terminal portion of α5 with supplementary interactions from α2 (Fig. 2a, b, d, f). The N-terminus of α5 is capped by the highly conserved Ser70 (see Supplementary Fig. 7 for a web logo representation of sequence conservation in the ProQ/FinO family). The N-terminus of this helix hydrogen bonds to all four non-bridging oxygens in two successive phosphate groups−C$_{21}$ and C$_{22}$. The C$_{21}$ phosphate is hydrogen bonded by Ser70 and Ser72, while the C$_{22}$ phosphate is hydrogen bonded by the main chain NHs of Lys71 and Ser72. Both non-bridging oxygens of the C$_{23}$ phosphate at the base of the stem are recognized by the side chains of Lys71 and Arg75. The G$_{20}$ phosphate is recognized by Lys73, as well as by Ser21 from α2 and residues in more N-terminal regions of α2 may make further long-range electrostatic contacts with the backbone of the RNA 5′ to this residue. Ser70, Lys71, and Arg75 are all well-conserved in the ProQ/FinO domain, while Ser72 is most commonly a Thr and Lys73 and Ser21 are less well-conserved (Supplementary Fig. 7). The precise recognition of all non-bridging oxygens in the three consecutive phosphates of C$_{21}$−C$_{22}$−C$_{23}$ along the same RNA strand presents an interesting possible mechanism for the recognition of an RNA duplex without direct interactions with both strands. We searched the protein−RNA and protein-DNA structure databases to uncover other examples of N-capped α-helices that recognize consecutive phosphates along a nucleic acid chain. We found that a similar mechanism of RNA recognition is used by the ROQ domain of the

mammalian Roquin protein[28,29], which binds three consecutive phosphates along a single strand of a hairpin RNA using a similar N-cap motif (Supplementary Fig. 8a, b). While many DNA binding proteins recognize DNA phosphates via hydrogen bonding interactions with α-helical N-terminal amide groups, we were unable to find any in which consecutive phosphates are recognized in a manner similar to either the ProQ/FinO or ROQ domains.

To test the hypothesis that the RocC helical N-cap motif could impose a specificity for hairpin-containing RNA partners compared to single-stranded RNA, we used ITC to compare the interactions of RocC$_{14-126}$ to either RocR$_{9bp-tet}$ or a ten nucleotide single-stranded RNA corresponding to just the RNA region directly in contact with RocC in the crystal structure (Fig. 3). The results confirmed a tight 1:1 interaction between RocC and RocR$_{9bp-tet}$ and furthermore revealed that the interaction is largely enthalpy-driven. RocC also bound the ssRNA with a 1:1 stoichiometry, albeit with an affinity that was ~19-fold weaker than for the stem-loop structure. In this case, the binding was still enthalpically-driven, however, the entropic cost of binding was much higher, consistent with the idea that the single-stranded RNA is able to make the same interactions with RocC, albeit with a higher entropic penalty due to the structural restraints imposed by the binding interaction on the flexible ssRNA.

## Recognition of the 3′ terminal nucleotide of RocR by a conserved β-hairpin-α-helix motif in the ProQ/FinO domain of RocC

The 3′ terminal nucleotide, U$_{28}$ bends back such that its 3′-hydroxyl hydrogen bonds with the penultimate phosphate linking U$_{26}$ and C$_{27}$ (Fig. 2a, c, d, f). The terminal U$_{28}$ is bound within a well-conserved and structurally rigid pocket that is formed between α5 and a β-turn at the N-terminus of α4. The pocket contains Tyr87 and Arg97, which are both among the most highly conserved residues in the ProQ/FinO domain family (Supplemental Fig. 7) and together recognize the

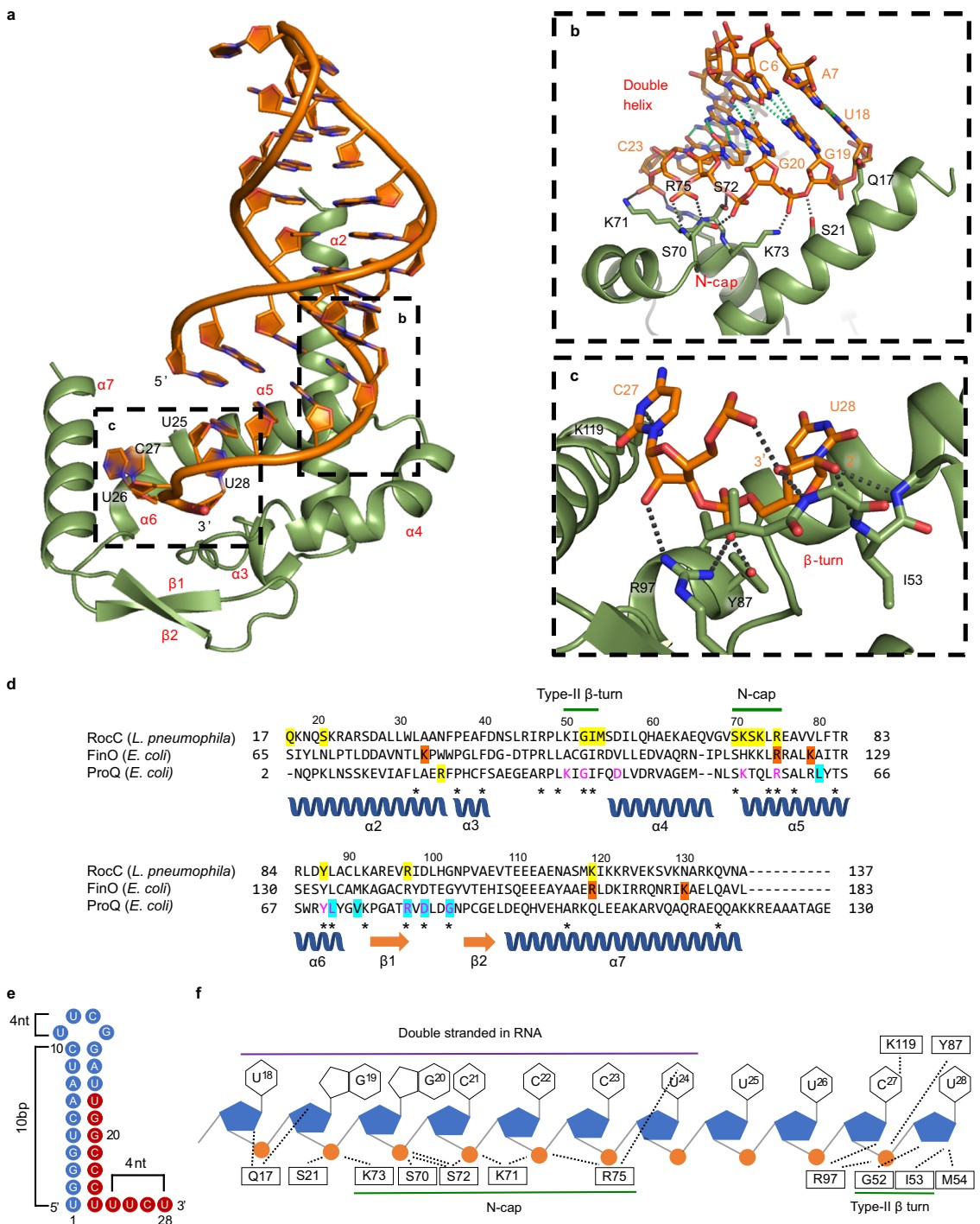

**Fig. 2 | Structural analysis of the ProQ/FinO domain of RocC bound to a RocR$_{SL3}$ variant. a** Crystal structure of RocC$_{14–126}$/RocR$_{9bp-tet}$ complex. Dotted boxes indicate the main interactions between protein–RNA. **b**, **c** are zoomed-in views from (**a**). Black dotted lines indicate hydrogen bonding between protein and RNA. Green dotted lines indicate hydrogen bonding for the base pairing. **d** Structure-based sequence alignment of the ProQ/FinO domains of RocC, FinO, and ProQ. Yellow highlights in RocC indicate residues in contact with RocR$_{9bp-tet}$ in the crystal structure. Asterisks indicate highly conserved residues across the three proteins. Orange highlights in the FinO sequence indicate residues that show strong cross-

linking with SLII of FinP[20]. Magenta letters in the ProQ sequence show residues that are critical for RNA binding in 3-hybrid screening[26]. Cyan highlights in ProQ indicate vital residues for ProQ function[30]. The indicated secondary structure is derived from the RocC/RocR crystal structure. **e** Schematic diagram of the RocR$_{9bp-tet}$ variant, which is crystallized with RocC. The red circles indicate the nucleotides in direct contact with the protein. **f** Schematic diagram of RocC-RocR interactions. The purple line indicates the region of double-stranded RNA structure; green lines indicate protein motifs in contact with RNA. Dotted lines show molecular interactions between protein and RNA.

phosphate of U$_{28}$. The β-turn-α-helix motif packs against the minor groove face of U$_{28}$, hydrogen bonding with the U$_{28}$ 2′ and 3′ hydroxyl groups. Gly52, which stabilizes the turn, is nearly completely conserved in the ProQ/FinO domain family. This interaction effectively

buries the 3′ end of the strand in the protein and provides a mechanism for the selective recognition of a 3′ terminal ribose sugar. We searched the protein–RNA structure database to find other examples of ribose recognition by similar β-turn-α-helix motifs (see "Methods"). The most

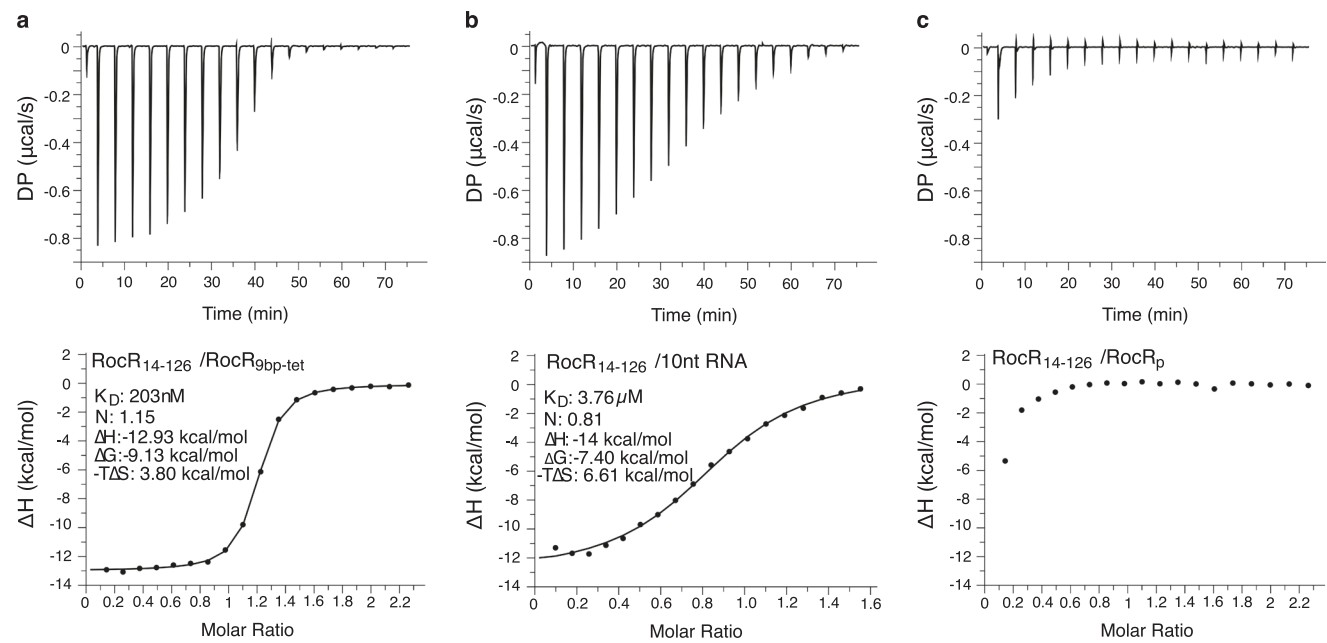

**Fig. 3 | Measurement of binding interactions between RocC$_{14-126}$ and either RocR$_{9bp-tet}$ or a single-stranded RNA. a** ITC analysis of RocC$_{14-126}$ with RocR$_{9bp-tet}$. **b** ITC analysis of RocC$_{14-126}$ with a 10 nucleotide single-stranded RNA. **c** ITC analysis of RocC$_{14-126}$ with RocR$_{SL3}$ containing a terminal 3'-phosphate (RocR$_P$). **a–c** Each experiment was repeated independently three times with similar results. Source data are provided as a Source Data file.

similar example we found was from the 40 S subunit of the eukaryotic ribosome where a β-turn-α-helix motif contacts a nucleotide within the 18 S rRNA in the same orientation with the same hydrogen bonding arrangement (Supplementary Fig. 8c, d).

The tight packing of the 3' nucleotide of RocR against RocC as well as the interaction of the 3'-hydroxyl with the penultimate phosphate suggests specific recognition of a terminal nucleotide with a 3'-hydroxyl group. To test this idea, we assessed the interactions of RocR$_{SL3}$ containing a terminal 3'-phosphate with RocC by ITC. In contrast to 3'-hydroxyl target RNAs, this RNA showed little if any binding to RocC, demonstrating the critical importance of the chemical structure of the terminal nucleotide for RocC recognition (Fig. 3c).

## Site-directed mutagenesis reveals the importance of specific RocC/RocR contacts for binding and DNA uptake in vivo

Guided by the RocC-RNA structure, we created a set of RocC mutants to test the contribution of individual amino acid–RNA contacts to the RocC/RocR interaction. In vitro, we assessed the binding affinity between RocR$_{9bp-tet}$ and each RocC$_{14-126}$ mutant via a fluorescence polarization (FP) assay (Fig. 4 and Supplementary Fig. 9). In vivo, the interaction between RocC and RocR is essential to stabilize RocR and promote the post-transcriptional repression of genes required for the uptake of DNA from the environment (transformation). Consequently, a strain in which RocC is absent or non-functional is more transformable than its WT counterpart ("hypercompetent" phenotype)[6]. We thus monitored the effects of different point mutations of RocR by testing the transformability of *L. pneumophila* strains expressing these variants compared to the WT protein (Fig. 4d).

Interestingly, the impacts of the different mutations on the binding affinity and the transformation efficiency are mostly in agreement between in vitro and in vivo experiments. Mutation of residues within and surrounding the 3' nucleotide-binding pocket yielded a significant impact on the RocC/RocR binding affinity and transformation efficiency. Mutation of the highly conserved Arg97 (R97M), which forms the base of the 3' nucleotide-binding pocket, resulted in an >200-fold reduction in binding affinity as well as a dramatic reduction in transformation repression similar to that observed

in the Δ*rocC* or Δ*rocR* controls (Fig. 4d). We mutated the absolutely conserved Tyr87 to phenylalanine (Y87F). In vivo, this mutation resulted in a complete loss of repression of transformation, however, in vitro this mutant was insoluble, suggesting a significant folding defect related to the mutation of this buried residue. Mutation of residues surrounding the 3' nucleotide-binding pocket showed more subtle effects. Mutations of residues on α7, N115A and K119D, showed less dramatic but still significant reductions (-sixfold) in binding affinity, but not in transformation repression. Asn115 is positioned within hydrogen bonding distance to the U$_{26}$ and/or C$_{27}$ base of RocR$_{SL3}$ and is most often a His or Tyr within the ProQ/FinO family (Supplementary Fig. 7). Lys119 is positioned to make a cation-π interaction with the U$_{26}$ base. However, within the ProQ/FinO family this residue is most often a Gln or Arg. Arg83 also lines the 3' nucleotide-binding pocket and is positioned to contact the 5'-most nucleotide. Mutation of this residue (R83D) led to a modest but statistically significant (-threefold) reduction in binding affinity, as well as a small reduction in transformation repression. Arg is commonly observed at this position in ProQ/FinO domains, however, Ser is the most conserved residue at this position (Supplementary Fig. 7). Mutation of two other residues in the pocket led to no significant reduction in binding affinity. Thr82 packs against the base of the terminal U$_{28}$ base, however, mutation of this residue to an alanine did not reduce binding or transformation repression, even though Thr is highly conserved at this position in ProQ/FinO domains. In addition, mutation of the highly conserved Ile51, which packs against the backbone of C27, to an alanine did not result in any reduction in binding and had no impact on transformation repression.

Mutation of residues in the N-cap RNA-binding motif also led to significant reductions in binding and biological activity. The most dramatic effect was observed for R75D which led to a >100-fold reduction in binding affinity, as well as a complete loss of the mutant protein's ability to repress transformation. K71D and K73D also displayed significant reductions in binding affinity (-12-fold) as well as an almost complete loss in transformation repression. S70A resulted in a less pronounced defect in binding and little if any effect on transformation repression, however mutation of both Ser70 and Ser72 to Ala resulted in stronger defects in binding and repression.

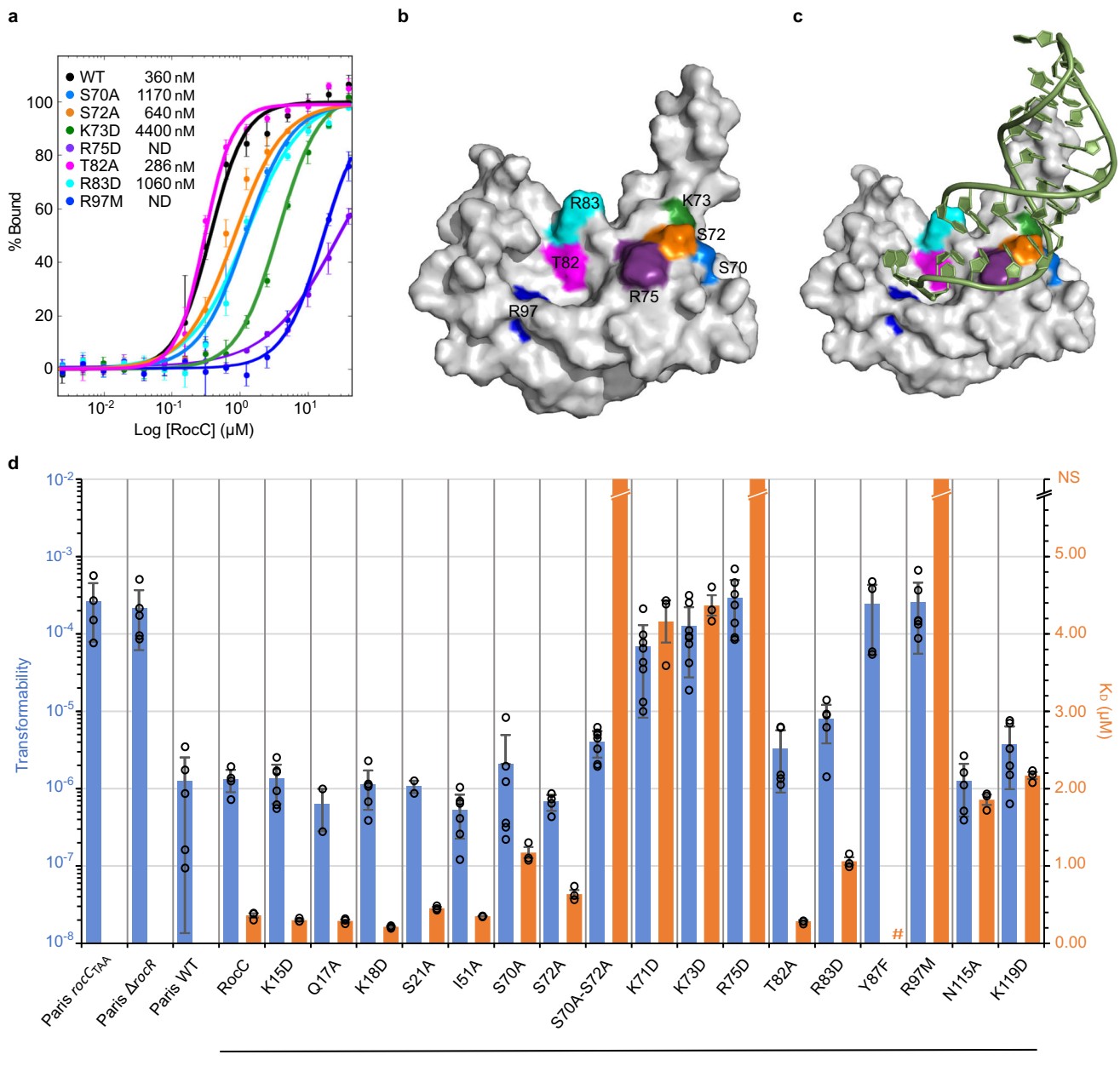

**Fig. 4 | Effects of RocC point mutations on RocR binding in vitro and trans-formability in vivo. a** FP binding assay for 5′ FAM-labeled RocR_SL3 with different RocC_{14–126} point mutants ($n = 3$). The error bars are standard error of the mean (SEM). **b, c** Surface representation of RocC_{14–126} alone (**b**) and with RocR (**c**) colored to indicate the positions of mutated residues. **d** A graph displaying RNA-binding affinities and transformation efficiencies for the indicated strains and RocC mutants are shown. Orange histograms indicate the $K_D$ values measured by FP ($n = 3$), and the blue histograms indicate the relative transformability of the indicated mutant or strain. Transformability is the ratio of the number of CFUs counted on selective medium divided by the number of CFUs counted on non-selective medium. # indicates a mutant which could not be purified due to low protein solubility. ND indicates mutants where RNA binding could not be detected. Values for the individual transformation measurements are shown (experiments were repeated at least twice on two independent clones), and the standard error of the mean (SEM) from three independent measurements are show for the FP binding data. Source data are provided as a Source Data file.

We also tested mutations in the N-terminal helix α2, which our structure suggested could make limited contacts to the RNA hairpin. Deletion of the N-terminal 13 residues, which are disordered in our structures, did not impact RNA-binding affinity or transformation repression. However further deletion to residue 24 resulted in a -threefold reduction in binding affinity, similar to what we had observed by EMSA, and a total loss of transformation repression (Fig. 1b, c). Individual mutation of positively charged residues in the 14–24 region (K15D and K18D) did not exhibit reduced RNA binding, nor did they cause significant defects in transformation repression

in vivo. Furthermore, we tested the binding affinity of a RocC construct containing an additional C-terminal predicted helical region (RocC_{1-137}). This construct had the same binding affinity as 1–126, indicating that this additional region does not play a significant role in RNA binding (Supplementary Fig. 9b, d).

In general, all the RocC mutants were detected in vivo however the three-point mutants with the strongest phenotypes (R75D, Y87F, and R97M) showed lower levels of protein compared to WT (Supplementary Fig. 10). This could be due to a structural effect of these particular mutations, and indeed, we find that the Y87F mutation likely

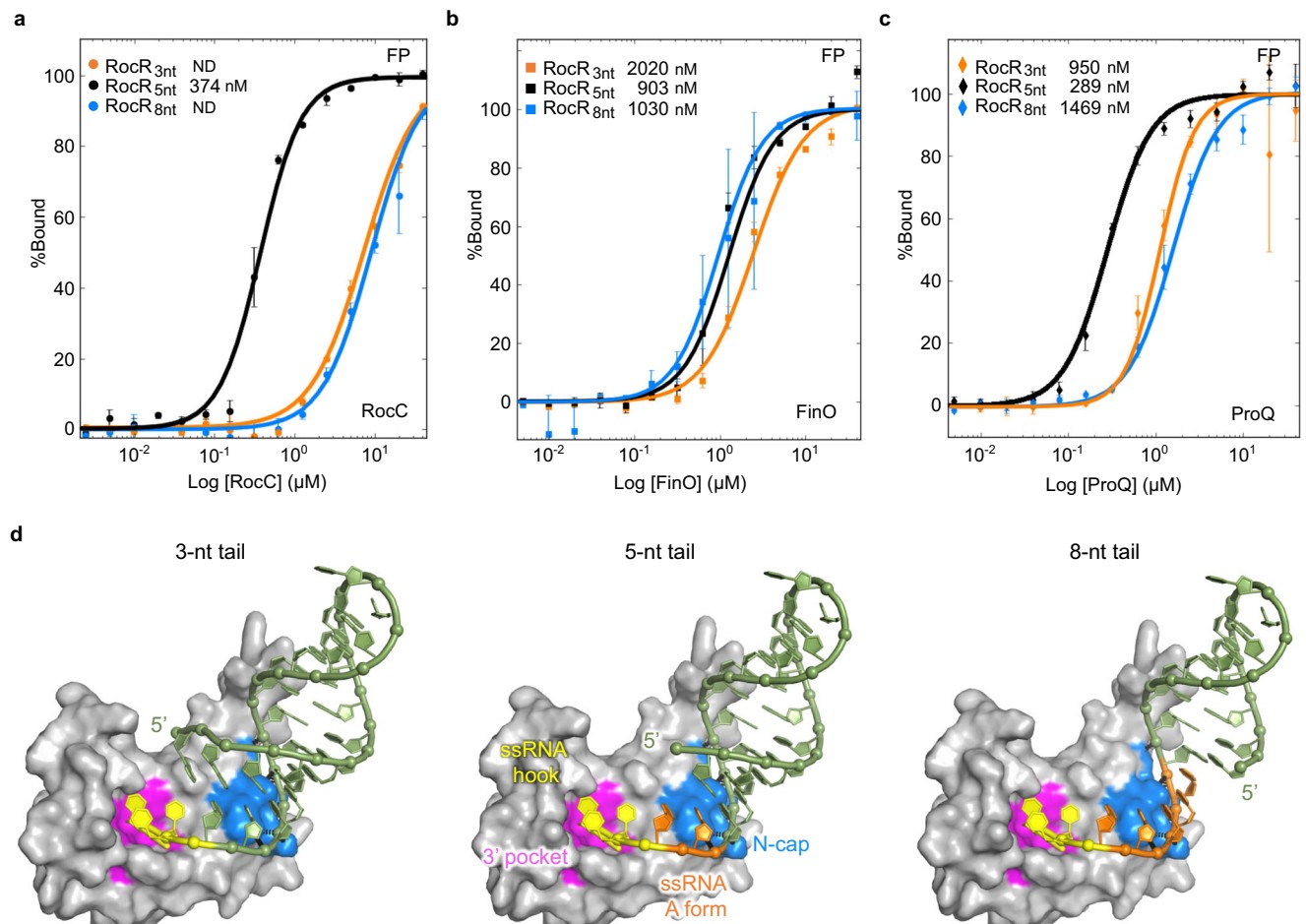

**Fig. 5 | Specificities for 3' tail length among FinO domain-containing proteins.** **a**–**c** FP binding assay for various tail lengths of 5' FAM-labeled RocR$_{SL3}$ with RocC$_{14-126}$ ($n = 3$). **a** With FinO$_{45-186}$ (**b**), or with ProQ$_{1-130}$. ND indicates mutants where RNA binding was too weak to determine a K$_D$. ND indicates mutants where RNA binding could not be detected. The error bars are the standard error of the mean (SEM). **d** Models for the interaction of RNAs with different 3' single-stranded tail lengths to RocC$_{14-126}$. Left panel: RNA with a 3-nucleotide tail. Center panel: RNA with a 5-nucleotide tail (as in RocR). Right panel: RNA with an 8-nucleotide tail. Source data are provided as a Source Data file.

does lead to a folding defect. However, the other mutants, R75D and R97M, in contrast are soluble in vitro. Because we also see less RocC in hypercompetent strains such as Δ*rocR*, we hypothesize that the RocC/RocR interaction might help stabilize both RocR and RocC and it was previously shown that ProQ mutants with impaired RNA binding were also less stable[30]. Thus the reduced levels of R75D and R97M might be due to their reduced interactions with RocR in vivo.

### ProQ/FinO domain proteins exhibit specificity for the length of the terminator 3' tail

ProQ/FinO family members exhibit profound differences in their ability to recognize different RNAs in vivo. Certain proteins, such as RocC and FinO, only have one biological partner whereas other family members, such as ProQ and NMB1681, can bind a range of RNAs. The structures of the ProQ/FinO domains of these proteins, in particular, their 3' nucleotide-binding pockets and N-cap motifs, are well-conserved between these proteins. Previous work demonstrated that FinO, ProQ, and NMB1681, like RocC, all can bind transcription terminator structures[6,16,21,22] and indeed, the sequence (Fig. 2d) and structural similarities of the ProQ/FinO domains of these proteins suggest they could bind to hairpin-3' tail RNAs very similarly to RocC. We hypothesized that part of the difference in the specificity of these different proteins might be their ability to bind RNAs with different lengths of 3' single-stranded tails. To test this, we created a set of

model terminators based on RocR$_{SL3}$ with 3-, 5-, or 8-nucleotide tails and measured the affinity of the ProQ/FinO domains of RocC, ProQ and FinO for these RNAs by FP (Fig. 5 and Supplementary Fig. 9c, e). All the ProQ/FinO domains bound at least one of the test RNAs with high affinity (K$_D$ < 1 μM). In agreement with the EMSA results, RocC showed a dramatic specificity for the 5-nucleotide tail, and only interacted weakly with the 3- or 8-nucleotide tails. Similarly, the ProQ/FinO domain of ProQ showed a strong binding to the 5-nucleotide tail RNA, and less binding to the other RNAs. In contrast, the ProQ/FinO domain of FinO showed less difference in binding specificity between the different RNAs, with only a weak preference for the 5- and 8-nucleotide tail lengths compared to the 3-nucleotide tail.

## Discussion
### RocC/RocR provides a model for the recognition of intrinsic transcriptional terminators by proteins with a ProQ/FinO domain

The structure of the RocC/RocR$_{SL3}$ complex presented here reveals a mechanism where RocC specifically binds its natural partner RocR by interacting with its terminator, i.e., the hairpin of SL3 and its 3' single-stranded polypyrimidine tail. The relevance of this structure to RocC/RocR binding in solution and in vivo is supported by extensive mutagenesis data. This structure also provides a model to understand RNA recognition by other ProQ/FinO domain proteins. The pocket that

binds the 3' nucleotide of the terminator is particularly well-conserved in the ProQ/FinO family. The position of the N-cap motif is absolutely conserved in the ProQ/FinO family, and Arg75 is almost completely conserved (80% identity) with the major substitution being lysine. The structure explains previous biochemical and mutagenesis data on FinO as well as ProQ (see Fig. 2d for an overview of the previous data). Site-specific protein–RNA cross-linking identified residues in and around the conserved 3' nucleotide-binding pocket (Lys125, Arg165), as well as the N-cap (Arg121) as key contact points for FinO[20]. Likewise, bacterial 3-hybrid experiments identified the same surfaces on the ProQ/FinO domain of ProQ as important to recognize transcriptional terminator structures[26]. Ribonuclease footprinting revealed dramatic protection of the FinP 3' tail by FinO, as well as protection of the first 3–4 nucleotides within the stem, precisely the same region that is bound by RocC in the RocC/RocR structure[17]. Interestingly, it was previously shown that the recognition of FinP by FinO is strongly dependent on the chemical structure of the 3' nucleotide[17], and we demonstrate here that phosphorylation of the 3'-nucleotide of RocR also abolishes RocC binding (Fig. 3). Thus, we suggest that RNA recognition by ProQ/FinO domain proteins likely will be specific for RNAs that terminate with a 3'-hydroxyl group. RNAs with alternative 3'-phosphate or 2',3' cyclic phosphate termini, which might arise as the result of ribonuclease digestion, would likely not be bound.

The ProQ/FinO family members that have been studied in detail (FinO, RocC, ProQ, and NMB1681) all bind transcriptional terminator structures but with varying degrees of specificity. While FinO and RocC bind to just one or two physiological partners, ProQ and NMB1681 bind to dozens of different sRNAs. Our structure of the RocC/RocR$_{SL3}$ complex suggests a mechanism for the specific recognition of the hairpin as well as for the 3' polypyrimidine tail. The duplex portion of the hairpin is recognized on one strand by the N-cap motif. Specificity for the duplex is likely conferred through recognition of the A-form geometry of the contiguous phosphate groups, rather than through a direct recognition of the 5' strand of the hairpin. This is supported by our finding that while an ssRNA can bind RocC, its binding is entropically disfavored compared to the hairpin form (Fig. 3). The lack of any direct interactions between the base pairs of the stem and RocC is consistent with the lack of apparent sequence specificity of recognition of this domain. We found that replacement of the G·C base pairs at the base of the stem only resulted in a small (~twofold) reduction in binding affinity, and previous results with FinO also indicate that this protein does not recognize specific sequences at this position[16]. The 3' polypyrimidine strand adopts a hook-like structure, stabilized by hydrogen bonding between the penultimate phosphate and the terminal 3'-hydroxyl (Fig. 2c). In this conformation, the strand tracks into the 3' nucleotide-binding pocket. While it is difficult to definitively model side chain−base hydrogen bonding interactions at this resolution, the structure does suggest some possibilities that may indicate specificity for pyrimidines within the tail. Highly conserved Arg75 participates in RNA backbone recognition, but may also hydrogen bond with the 5'-most uridines of the polypyrimidine tail. The next 2 nucleotides are packed against α7 in a manner that may sterically favor pyrimidines over purines, and residues such as Lys119 and Asn115 may provide sequence specificity through hydrogen bonding interactions with these bases. Consistent with this idea, mutations at these residues reduce binding affinity, however, transformation repression efficiency is not affected. The uridine base of the terminal 3' nucleotide packs against Thr82, however mutation of this residue does not reduce binding affinity or transformation repression efficiency.

Our results also suggest that different ProQ/FinO domains have different specificities for the length of the 3' ssRNA tail (Fig. 5). Biologically, both FinO and RocC bind RNAs with 5-nucleotide ssRNA tails, while ProQ tends to bind sRNA targets with shorter 4-nucleotide tails[13,25]. Biochemically, we find that the isolated ProQ/FinO domains of RocC, FinO, and ProQ all have differing tail length specificities. RocC

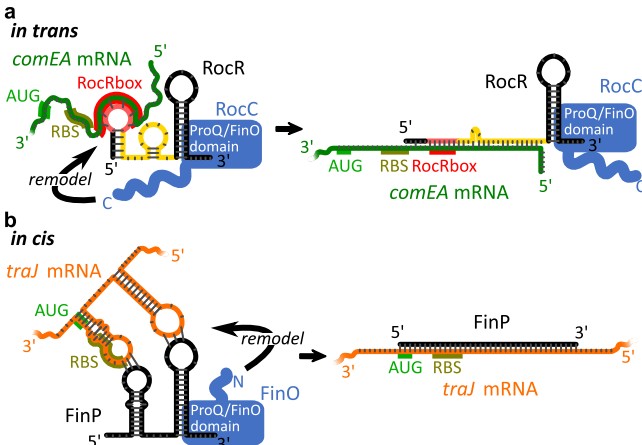

**Fig. 6 | ProQ/FinO domain proteins bind transcriptional terminator structures to regulate RNA-RNA interactions. a** Many ProQ/FinO domain proteins, such as RocC, facilitate in trans RNA association between sRNAs and target mRNAs. The ProQ/FinO domain specifically binds the transcriptional terminator of the RocR sRNA, stabilizing it against degradation. Key to RNA-RNA association is the recognition of the sRNA seed sequence with its complementary region (the RocR box) in the target mRNA. The C-terminal region of RocC is required to facilitate the recognition and translational repression of target mRNAs such as comEA. **b** Many plasmid-encoded ProQ/FinO domain proteins, such as FinO, regulate in cis sense-antisense RNA interactions. Similar to RocC, the ProQ/FinO domain of FinO also stabilizes the antisense RNA FinP against degradation. Initial RNA-RNA interactions are thought to involve loop–loop kissing interactions, which then proceed to duplex formation between the two RNAs, resulting in translational repression of the traJ mRNA target. In this case, the flexible N-terminal region of FinO is thought to be key in facilitating RNA-RNA interactions.

binds very specifically to its 5-nucleotide-tailed RNA target. ProQ also preferentially bound to the 5-nucleotide-tailed RocR compared to versions with either 3- or 8-nucleotide tails, while FinO showed little preference for the different tail lengths. Our structure would predict a minimal ssRNA tail length of three nucleotides, corresponding to the portion of the tail which must contour into the nucleotide-binding pocket (Fig. 5d). In our structure, the 5' nucleotide forms a U−U mispair with the first nucleotide of the 3' polypyrimidine tract, where the 5' nucleotide is packed against α5, however, it is unclear if this mispair would be possible in the case of a longer RNA extended at the 5' end. It is possible that different ProQ/FinO domain proteins all bind the 3' strand in a way that is similar to what is observed in the RocC/RocR complex, however differences in how much the 5' region can pair with the 3' region might be determined by interactions with residues from α5 and possibly α7 (Fig. 5d).

## Implications of the RocC-RocR structure for RNA remodeling

One of the most intriguing properties of ProQ/FinO domain RNA chaperones is their ability to directly facilitate RNA-RNA association. This has been best studied for FinO, where the protein has been demonstrated to facilitate sense-antisense pairing between FinP and traJ mRNA in vitro[9,19,31–33] (Fig. 6). FinO can also enhance interactions between the minimal transcriptional terminator SLII of FinP and its complementary region of traJ and it can also catalyze strand exchange between duplex and ssRNAs, implying that FinO can destabilize the double-stranded nature of the bound SLII hairpin. Sense-antisense recognition is thought to proceed via kissing interactions between complementary loop regions of the RNAs with the subsequent unwinding of internal hairpins and inter-strand duplexing[5]. However, how FinO could at once bind to a hairpin RNA while destabilizing its base pairing was unclear. The RocC/RocR structure reveals that RocC only binds the 3' strand of the hairpin, leaving the 5' side unencumbered so that it could be peeled away to allow duplexing with a complementary RNA.

The exact role of other ProQ/FinO domain proteins in facilitating RNA-RNA association is less clear. Many of these proteins, such as RocC and ProQ, mediate sRNA–target RNA recognition where there is only limited base pairing between the two RNAs (Fig. 6). Most ProQ/FinO domain RNA partners identified to date are highly structured, and it is intriguing to speculate that the intrinsically disordered regions outside of the ProQ/FinO domain may be critical for chaperone activity. This hypothesis is supported by the observation that both the FinO_N domain of FinO (N-terminal to the ProQ/FinO domain) and the C-terminal domain of RocC (C-terminal to the ProQ/FinO domain) were shown to be essential to their function. Deletion of the flexible FinO_N domain abrogated duplexing and strand exchange activities but did not reduce RNA-binding activity of FinO. This mutant was also unable to repress conjugation which suggests that this domain is critical for the RNA chaperone activities of FinO in vivo[19]. Protein–RNA cross-linking and FRET studies suggest that this region directly contacts RNA[20]. In a similar manner, deletion of the C-terminal domain of RocC does not impair its binding to RocR or RocR stability but causes a loss of the post-transcriptional repression of the mRNA targets[6]. ProQ also contains a C-terminal region containing a Tudor domain which contacts RNA[24] and facilitates RNA strand exchange and duplexing[21] (Fig. 6). More studies are needed to better understand the potential dynamic interactions between the ProQ/FinO domain, the associated domains, and their RNA partners and targets.

## Methods

### Bacterial strains and growth conditions

The *L. pneumophila* strains in this study are derived from the Paris clinical isolate (Outbreak isolate CIP107629). These strains (see genotypes in Supplementary Table 2 and construction details below) were grown in liquid media ACES [N-(2-acetamido)-2-aminoethanesulfonic acid]-buffered yeast extract (AYE) or on solid media ACES-buffered charcoal yeast extract (CYE) plates at 30 °C or 37 °C. Liquid cultures were performed in 13-mL tube containing 3 mL of medium in a shaking incubator at 200 rpm. When appropriate, kanamycin and streptomycin were used at 15 µg mL$^{-1}$ or 50 µg mL$^{-1}$, respectively.

*Escherichia coli* strains (see Supplementary Table 2) were cultivated in LB medium with shaking or on LB-agar plates at 37 °C. When appropriate, kanamycin and ampicillin were used at 50 µg mL$^{-1}$ and 100 µg mL$^{-1}$, respectively.

### Protein expression and purification

For each RocC* (GenBank ID: CAH11296.1), a single colony of BL21-Gold (DE3) bearing one of the pGEX-6P-1_RocC* plasmid was inoculated into 25 mL LB with 100 µg mL$^{-1}$ ampicillin and 35 µg mL$^{-1}$ kanamycin and incubated at 37 °C for 18 h with shaking. This preculture was then inoculated into 1 L LB with 100 µg mL$^{-1}$ ampicillin and 35 µg mL$^{-1}$ kanamycin and grown to O.D. 0.6–0.8 at 37 °C. GST-PP-RocC* expression was induced with 0.3 mM IPTG and incubation was continued at 18 °C for 20 h. The cell pellet was collected by centrifugation, flash frozen and stored at −80 °C. The cell pellet was resuspended in lysis buffer (50 mM HEPES pH 7.3, 500 mM NaCl, 5% glycerol, 1 mM DTT) and lysed using an Emulsiflex-C3 high-pressure homogenizer (Avestin). The lysate was incubated with 10 mL glutathione beads for 1 h at 4 °C. Incubated beads were washed with 150 mL of lysis buffer and the GST tag was removed by digestion with 3 C protease for 18 h at 4 °C. The digested protein was collected and purified by Superdex 75 16/60 gel filtration chromatography in different buffers depending on the purpose (ITC/EMSA/FP binding assay buffer: 25 mM pH 7.3 HEPES, 150 mM NaCl, 10% glycerol; apo-protein crystallization buffer: 25 mM Tris pH 8.0, 30 mM NaCl, 1 mM DTT; complex crystallization buffer: 10 mM HEPES-KOH pH 7.5, 100 mM KCl, 5 mM MgCl$_2$, 1 mM TCEP). FinO (45–186)[19] and ProQ (1–130)[21] were purified in the same way as RocC*.

### RNA synthesis, expression, and purification

RNAs were prepared in three ways for different analyses. RNAs for EMSA were produced using in vitro transcription (see Supplementary Table 4), purified using denaturing PAGE, and radiolabeled as previously described[34]. RNAs for NMR spectroscopic applications were produced by solid phase synthesis and $^{15}$N labeled nucleotides for assignment purposes were incorporated as described earlier[35]. FAM-labeled RNAs for FP and RNAs for ITC were either purchased from IDT or synthesized. RNAs for crystallization were prepared using anion exchange followed by gel filtration, as previously described[34].

### Crystallization and crystallographic data collection

In all, 30 mg mL$^{-1}$ of RocC$_{24-126}$ in 25 mM Tris:HCl pH 8.0, 30 mM NaCl, 1 mM DTT was crystallized by hanging drop vapor diffusion with a reservoir solution (0.2 M ammonium acetate, 0.1 M HEPES pH 7.5, 10% PEG 3350) at 16 °C. Protein/reservoir solution ratios of 1:1 or 1:2 yielded crystals. 32 mg mL$^{-1}$ of RocC$_{1-126}$ in 25 mM Tris:HCl pH 8.0, 30 mM NaCl, 1 mM DTT was crystallized by hanging drop vapor diffusion with a reservoir solution consisting of 0.2 M ammonium sulfate, 0.1 M HEPES:NaOH, pH 7.3, 25% PEG 3350 at 16 °C in either ratio 1:1 or 1:2 of protein solution:resevoir solution. The quality of RocC$_{1-126}$ crystals was improved using additive screens (HR2-428, Hampton research) at 4 °C. In all, 2 µL of 32 mg mL$^{-1}$ RocC$_{1-126}$ was mixed with 1.6 µL of reservoir solution (0.2 M ammonium sulfate, 0.1 M HEPES:NaOH, pH 7.3, 25% PEG 3350) and 0.4 µL of the additive screen was mixed to make a 4 µL drop. Various additives produced high quality of crystals: multivalent ions (0.1 M barium chloride dihydrate, 0.1 M strontium chloride hexahydrate, 0.1 M yttrium(III) chloride hexahydrate, 0.1 M chromium (III) chloride hexahydrate), Linker (0.3 M glycyl-glycyl-glycine), polymer (10% w/v polyethylene glycol 3350), carbohydrate (30% sucrose, 12% w/v myo-Inositol), organic, non-volatile solvent (30% w/v 1,6-hexanedio), organic, volatile solvents (40% v/v tert-butanol, 40% v/v 1,3-propanediol). Homogenous RocC$_{14-126}$: RocR$_{9bp-tet}$ complex in 10 mM HEPES-KOH pH 7.5, 100 mM KCl, 5 mM MgCl$_2$, 1 mM TCEP was purified from excess free components using gel filtration chromatography and was crystallized by sitting drop vapor diffusion against a reservoir buffer containing 0.2 M lithium sulfate 0.1 M Tris: HCl, pH 8.5 30% (w/v) PEG 4000 (Top96, A10, Anatrace). Crystals grew at 1:1 protein:precipitant ratio at room temperature.

Data for RocC$_{24-126}$ were collected on a RIGAKU MICROMAX-007 HF with a DECTRIS PILATUS3 R 200K-A detector to a final resolution of 2.10 Å. Data for RocC$_{1-126}$ were collected at CLS (Canadian Light Source) BEAMLINE 08ID-1 with DECTRIS PILATUS3 S 6 M detector to a resolution of 2.02 Å. Crystals of the RocC$_{14-126}$/RocR$_{9bp-tet}$ complex were collected at ALS (Advanced Light Source) BEAMLINE 8.2.2 with ADSC QUANTUM 315r detector and data was obtained to a resolution of 3.20 Å. All data was integrated with HKL-2000[36].

### Crystallographic structure determination and refinement

The structure of RocC$_{24-126}$ was determined by molecular replacement (MR) using Phaser[37]. A pruned model of FinO$_{85-158}$ generated using sculptor[38] was used as a search model. Two protomers were found in the asymmetric unit. Non-crystallographic symmetry (NCS) was used in the early stages of refinement using PHENIX[39]. Missing parts of the model were manually built in Coot[40] and refined using the PHENIX to a R$_{work}$ of 16.1% and an R$_{free}$ of 23.8% to a final resolution of 2.10 Å.

The structure of RocC$_{1-126}$ was solved by MR, using RocC$_{24-126}$ as a search model with Phaser[37]. Nine protomers were placed in the asymmetric unit. NCS was used throughout the refinement and revealed α-helical density for the N-terminus of RocC for two of the protomers. Manual building and refinement were carried out with Coot[40] and PHENIX[39] with the assistance of ninefold NCS to a final R$_{work}$ of 19.3% and an R$_{free}$ of 21.3% to a resolution 2.02 Å.

The RocC$_{14\text{-}126}$: RocR$_{9bp\text{-}tet}$ complex was phased by molecular replacement using the high-resolution structure of RocC$_{1\text{-}126}$ as a search model. Ten protomers were placed in the asymmetric unit and NCS was used throughout the refinement. Refinement of ten protomers showed different densities indicative of RNA bound to four of the ten protomers. This difference density map was improved through fourfold NCS averaging (Supplementary Fig. 5). An ideal A-form helix was then fit to the stem portion of the four RNA molecules. The hairpin loops were built using a related tetraloop model with PDB ID: 4Z3S [https://www.wwpdb.org/pdb?id=pdb_00004z3s]. Comparison of difference maps at low vs high sigma cutoffs helped to distinguish phosphate groups from sugar and base moieties, which was particularly helpful in the building of the 3′ single-stranded tail. The structure was refined using non-crystallographic symmetry restraints in PHENIX[39] to a final R$_{work}$/R$_{free}$ of 21.9% and 27.1%, respectively, to a final resolution of 3.20 Å (see Supplementary Table 1 for crystallographic statistics).

### Electrophoretic mobility shift assay (EMSA)

In vitro transcribed RNAs were 5′ radiolabeled with ATP, [γ-$^{32}$P] (PerkinElmer) using T4 polynucleotide kinase (Invitrogen) and purified using denaturing PAGE as described[34]. Labeled RNAs were incubated 30 min on ice with proteins from 0 to 32 µM concentration in a final volume of 5 µL EMSA reaction buffer (25 mM HEPES pH 7.3, 150 mM NaCl, 4 mM MgCl$_2$, 10% glycerol, 1 mM DTT, 0.5 mg mL$^{-1}$ yeast tRNA (ThermoFisher), and 12 U RNaseOUT (ThermoFisher)). The reactions were mixed with 5x native gel loading dye (10 mM Tris pH 8.0, 50% glycerol, 0.001% bromophenol blue, 0.001% Xylene cyanol FF) and separated on 10% native gels run in 1x Tris-Glycine buffer at 4 °C. Gels were dried and imaged with phosphor imaging screens (Molecular Dynamics). Band intensities were quantified using Molecular Dynamics ImageQuaNT TL software (GE Healthcare). Prism5 (GraphPad) was used for graph fitting and equilibrium dissociation constant (K$_D$) calculation with one site-specific binding with Hill slope:

$$Y = \frac{B_{max} \times X^h}{(K_D{}^h + X^h)} \tag{1}$$

where X is the protein concentration, Y is the fraction bound, B$_{max}$ is the maximum specific binding, and h is the Hill slope. The input data are X and Y, and B$_{max}$, h, and K$_D$ are obtained from the fitting procedure. All EMSA results were repeated at least three times.

### Isothermal calorimetry assay (ITC)

ITC was performed using MicroCal PEAQ-ITC (Malvern panalytical) at 25 °C. Ten nucleotide RNA (5′-GGCCCUUUCU-3′) was purchased from IDT and SL3-9bp-tet was chemically synthesized. RocC$_{14\text{-}126}$ was prepared as described in the protein expression and purification section. All components were dialyzed in ITC buffer (25 mM HEPES pH 7.3, 150 mM NaCl, 10% glycerol, 1 mM TCEP). In all, 30–50 µM of protein in the sample cell was titrated with RNA titrand at concentrations between 350 and 400 µM. In total, 2 µL (except for first injection) of concentrated titrand were injected 19 times every 240 s. Data were analyzed using MicroCal PEAQ-ITC Analysis Software (Malvern analytical).

### Fluorescence polarization assay (FP)

5′ FAM-labeled RocR$_{SL3}$ was purchased from IDT and 5′ FAM-labeled RocR$_{3nt}$, RocR$_{8nt}$ were chemically synthesized. 5 µL of 80 nM RNA in FP reaction buffer (25 mM HEPES pH 7.3, 150 mM NaCl, 4 mM MgCl$_2$, 10% glycerol, 1 mM DTT, 0.4 mg mL$^{-1}$ yeast tRNA (ThermoFisher)) was mixed with 15 µL of point mutated RocC$_{14\text{-}126}$ to a final protein concentration of between 320 µM and 2.4 nM in a 20 µL reaction. Reactions were transferred to a 384-well plate and were incubated for 1.5 h

at 25 °C. FP experiments were conducted using an Wallace EnVision manager (PerkinElmer) using 485 nm excitation and recorded at 538 nm. All experiments were repeated three times and the dissociation constant was calculated by fitting results to a sigmoidal curve with a 4-parameter logistic (4PL) equation:

$$y = d + \frac{(a-d)}{(1 + \frac{x}{c})^b} \tag{2}$$

where x is the protein concentration, y is the percent bound, a and d are the estimated minimum and maximum values. b is the slope factor and c is the protein concentration that gives 50% binding. Under these conditions (protein concentration >> RNA concentration), c ≅ K$_D$. Input data is x and y and a, b, c, and d are all obtained from the 4PL fitting.

### Transformability assay

Transformability of *L. pneumophila* strains was assessed as previously described using conditions in which the WT strain is not transformable but transformation is highly efficient in hypercompetent mutant (such as the *rocC$_{TAA}$* strain, which lacks RocC)[6].

The strains were streaked on CYE solid medium from the −80 °C frozen stock culture and incubated 3 days at 37 °C. The strains were then re-streaked on a new CYE plate and incubated overnight at 37 °C to obtain exponentially growing cells. Bacteria were resuspended to an OD$_{600}$ = 1 (≈10$^9$ cells mL$^{-1}$) in 3 mL AYE. Two times 1 mL of cell suspension were centrifuged for 3 min at 5000 × $g$ in a table-top microcentrifuge, and each pellet was resuspended in 50 µL of AYE with or without transforming DNA. Each suspension was spotted on a new CYE plate and let to dry. The plate was incubated overnight at 37 °C. Each spot was resuspended in 200 µL AYE. Ten-fold serial dilutions were then plated on non-selective medium and selective medium. Plates were incubated for 72 h at 37 °C and colony-forming units (CFU) counting was performed. Transformation frequency is the ratio of the number of CFUs counted on selective medium divided by the number of CFUs counted on non-selective medium. To test the transformability of the RocC punctual mutants (stains KanR), the DNA used was 2 µg (for 10$^9$ cells) of a 4-kb PCR fragment centered on a mutated allele of the *rpsL* gene. Transformants were thus selected on CYE + Streptomycin. To test the transformability of the RocCΔN mutants, the DNA used was either the same *rpsL*$^R$ PCR as above or 2 µg (for 10$^9$ cells) of the pGEM-ihfB::Kan, a plasmid containing a kanamycin-resistance cassette (KanR) inserted in the *ihfB* gene of *L. pneumophila*. As this plasmid is non-replicative in *L. pneumophila*, the internalized molecules recombine with the chromosome via a double crossover allowing the integration of the KanR cassette in the *ihfB* locus. Transformants were thus selected on CYE + Streptomycin or CYE + Kanamycin. For each tested mutants, transformation assays were performed three to seven times.

### Reporting summary

Further information on research design is available in the Nature Portfolio Reporting Summary linked to this article.

## Data availability

The determined structures in this study have been deposited in the Protein Data Bank under accession code 7RGS (RocC24-126), 7RGT (RocC1-126), and 7RGU (RocC14-126/RocR9bp-tet). All data supporting the findings in this study are available upon reasonable request from the corresponding author(s). Source data are provided with this paper.

## Code availability

The Perl scripts used to search the PDB for protein/RNA interaction motifs have been deposited in Github under the following link https://github.com/Glover-Lab/Protein-RNA-interaction-motifs.

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

## Acknowledgements

Work in the JNMG laboratory is supported by the National Sciences and Engineering Council of Canada (NSERC Discovery Grant RGPIN-2016-05163) and the Canadian Institutes of Health Research (CIHR168972). Work in X.C. laboratory was performed within the framework of the LABEX ECOFECT (ANR-11-LABX-0048) of Université de Lyon, within the program "Investissements d'Avenir" (ANR-11-IDEX-0007) operated by the French National Research Agency. This work was supported by a grant from Agence Nationale de la Recherche to L.A. (Project RNAchap, ANR-17-CE11-0009-01). Work in the AMM laboratory is supported by the Natural Sciences and Engineering Research Council of Canada (NSERC Discovery Grant RGPIN-2016-05175). Work in the C.K. laboratory is

supported by the Austrian Science Fund (FWF, P32773, P34370). Work in the MT laboratory is supported by the Austrian Science Fund (FWF, P33953). C.K. and M.T. acknowledge funding by the Austrian Research Promotion Agency FFG (West Austrian BioNMR, 858017).

## Author contributions

H.J.K., L.A., and J.N.M.G. designed research; C.K. and M.T. provided reagents, D.K. performed RNA synthesis; D.K., R.Ei., and R.Z. performed NMR and ITC experiments; R.P. performed site-directed mutagenesis; H.J.K. and M.B. purified protein and RNA and carried out EMSA and FP RNA-binding experiments with additional assistance from R.P., S.P., J.S., and A.R.O.; H.J.K. and R.A.Ed. determined crystal structures of RocC and complexes with RNA; L.A. carried out competence assays with additional assistance fom F.P.-F.; H.J.K., L.A., M.B., A.M.M., C.K., M.T., X.C., and J.N.M.G. analyzed the data; L.A., X.C., C.K., M.T., and J.N.M.G. provided funding; and H.J.K. and J.N.M.G. wrote the paper with contribution from L.A., and all other authors.

## Competing interests

The authors declare no competing interests.
