## [Peer Review File · Nature Communications]

Structural basis for recognition of transcriptional terminator structures by ProQ/FinO domain RNA chaperonesREVIEWER COMMENTS

Reviewer #1 (Remarks to the Author):

In this study, Kim et al. report the structure of the ProQ/FinO-domain protein RocC from *Legionella pneumophila* along with part of the bound small RNA RocR. The authors test predictions from their structure by assaying RNA binding of mutant proteins in vitro (via fluorescence polarization) and complementation in vivo (through the ability of mutant strains to take up DNA). The authors conclude the 3' nucleotide of the terminator is recognized by a conserved pocket in RocC, potentially explaining the differences in 3' tail recognition by other ProQ/FinO family proteins.

The data are complete and clearly presented and contribute to our understanding of this important family of RNA binding proteins. I only have relatively minor comments:

Page 10: Why did the authors not examine the consequences of mutating Gly52 given its conservation and proposed role in interacting with the 3' end?

Figure 2b: I thought the double helix was very difficult to see in the figure and am hoping this can be improved by changing the colors/shading. There should be a key for the colors in these structure panels.

Figure 4d (and line 259): The authors need to be explicit about how they are expressing the mutant RocC derivatives in *L. pneumophila*. Are all of the mutant proteins present at the same levels?

Figure 6: While this figure provides a review of the function of ProQ/FinO-domain proteins, it does not summarize what is discovered in this study. I suggest the authors replace this figure and the description of it with more discussion of how the different ProQ/FinO-domain proteins bind tails of different lengths, possibly with a model illustrating the findings regarding the length specificity.

Reviewer #2 (Remarks to the Author):

The group of RNA binding proteins containing a ProQ/FinO domain have been recognised as an important class of RNA chaperones and mediators of regulatory RNA mediated control of gene expression in several bacterial species. How the proteins interact with RNA is not known precisely, and the work by Kim et al has now illuminated the atomistic details and elaborated rules for recognition. Using x-ray crystallography together with biophysical methods to test the observed contacts, the authors establish that a stem loop structure and 3' single stranded region with defined length and with 3'OH group are key determinants for the action of *Legionella pneumophila* RocC and its related homologs. The in vivo function is corroborated in an assay for transformability. The experimental work is well done and the paper is well written and the results explained clearly and insightfully. There are a few minor comments that will hopefully be helpful to for the authors to consider:

It may be helpful to explain whether the full length RocC protein contains other domains, even if these are intrinsically disordered, and whether these could potentially cooperate with the ProQ/FinO domain.

An analogous interaction of the helical N-cap interaction with phosphate occurs for phosphate ion interaction at the active site of the exoribonuclease polynucleotide phosphorylase (as well as aracheal homolog). It might be helpful to comment on this.

Line 176 The asymmetric unit contains 6 copies of apo-RoC14-126 and 4 copies of the protein-RNA complex. Are there any higher order arrangements with additional interactions of apo-RoC14-126 with the RocR9bp-tet RNA that could support higher order assembly?

It might be that RocC will not bind RNA if the 3' end has a 2',3' cyclic phosphate. Is such a species likely to arise in any processing step of RNA targets?

In Figure 4, the Kds and transformability do not seem to scale in any simple way (compare for example S70A-S72A with K71D and K73D). Can the authors please comment on this?

Line 141 here after -> hereafter ?

Figure 1a – here and elsewhere, is the inset for confidence for prediction necessary, since it is 1 throughout? Same for other figures too?

Figure 2, label secondary structural elements in panel a to match labels in panel d?

Reviewer #3 (Remarks to the Author):

The manuscript "Structural basis for recognition of transcriptional terminator structures by ProQ/FinO domain RNA chaperones" by Kim et al. provides important structural, biochemical, and functional data about the way the *Legionella* ProQ homologue RocC recognises its target sRNA and represses the competence regulon in this bacterium. The ProQ/FinO family of RNA chaperones are widespread among gamma-, beta- and alpha-proteobacteria, where they play diverse roles ranging from narrow-range plasmid conjugation control to competence regulation to global post-transcriptional gene expression programming. While the biochemical basis and regulatory rationale of ProQ-like chaperones have been extensively studied, and their various modes of action are getting relatively well understood, some critical aspects of ProQ-RNA interactions remain elusive. In particular, it has not been known how ProQ-like chaperones can sense the structure of their RNA ligands and 'measure' the length of the 3'-terminal U-rich tail, which are two critical features of their RNA recognition mechanism distinguishing them from another major type of bacterial chaperones, Hfq. By combining X-ray crystallography, biochemical and genetic assays, the authors convincingly show that RocC, interacting nearly exclusively with the sugar-phosphate backbone on one side of the target RNA, can read its A-helical shape and specifically recognise, by enforcing an unusually distorted conformation of a short, 4-5-nucleotide-long single-stranded tail, its 3'-terminal OH-group. The authors demonstrate that this binding mode is critical for RocC function in vivo. Finally, they explore the effect of the 3'-terminal single-stranded tail on RNA binding by two other major representatives of the family, FinO and enterobacterial ProQ. These experiments bring about better understanding of how these proteins can select their targets among a wide variety of available ligands. The authors have generated an impressive amount of high-quality data, and their findings are extremely valuable to the field: this structure is truly a game-changer in our knowledge of how ProQ-like proteins engage their targets. I would only propose a couple of minor changes which, however, will certainly be required to bring the manuscript to a publishable level.

1. The authors rapidly narrow down their attention to SL3, based on the affinity argument and for the sake of easier crystallisation. However, similar binding strength shown by RocR and SL3 alone does not necessarily mean that other parts of either molecule (RocR or RocC) do not participate in the interaction. A relevant example is Hfq-dependent sRNAs, which interact with Hfq primarily via the proximal face, and this binding is critical for their stability and largely defines affinity. However, nearly all sRNAs have additional binding regions engaging either the rim or the distal face of Hfq. These interactions do not contribute much to affinity but play a vital role in their functionality. I think the authors should be cautious about such possible multi-site interactions (e.g. involving the C-terminal domain of RocC and SL1 or SL2). While the binding mode they describe in this manuscript is certainly the most important one, it is likely not the only one in the context of the full-size molecules.

2. Looking at Figure 2b, I wonder whether having a longer helix 2 (continuing more toward the original N-terminus) could enable additional contacts between RocC and the base-paired portion of

- SL3. What do authors think about it? Can such an interaction, if any, be modelled?
3. Ll. 32-33: More correct would be "bound to the transcriptional terminator of its primary partner, the sRNA RocR".
 4. Ll. 67-68: sRNAs control not only stress responses but many other things too (e.g. competence), and this should be reflected in this sentence. Moreover, not all of them work by base-pairing with mRNA targets (although most certainly do).
 5. L. 79: "has enabled the elucidation".
 6. Ll. 80-83: The unusual situation described in these two sentences is in a way unique to Salmonella Typhimurium strain SL1344, which has three plasmids with a somewhat entangled relationship between their FinO-like regulators and their sRNA ligands. I think it is important to make it clear and probably better explain what is going on in this specific case (for the sake of clarity).
 7. L. 162: The reference to Figure 1c is not suitable here.
 8. L. 167: Should be "Figure 1d..."
 9. L. 172: The sentence is ambiguous. If I got it right, I would insert a comma here: "using gel filtration chromatography, and SEC-MALS confirmed..."
 10. Ll. 283-285, 376-378, Fig. 4d: I think the authors overstate the effect of the T82A mutation on transformability. It might be statistically significant, but it has certainly not a big size effect. This also agrees with the affinity measurement.
 11. L. 314: Unless I am mistaken, the authors do not show results of this modelling. Do they refer to the published data cited just before in the same sentence?
 12. Section "Protein expression and purification": I suppose the authors meant "100 µg/ml" and "35 µg/ml" for antibiotics concentration.
 13. Same section, L. 510: It is not clear what "NaCl 3 0mM" exactly means.
 14. Ll. 595-596: Please, clarify/correct "Molecular Dynamics ImageQuaNT TL software".
 15. For the two equations in Materials and Methods (ll. 598-599 and ll. 622-623), it would be useful to provide definitions (and units) for all symbols in more canonical, biochemical terms. For instance, the 4PL equation for FP data remains quite cryptic with respect to what one should measure or fit and how one calculates K_d values in the end.
 16. Ll. 618-619: "conducted in/on an Envision 2103 multilabel plate reader"?
 17. Section "Western blot analysis of RocC and RocC mutants". Unless I am mistaken, western blotting data are not shown anywhere in the manuscript. Does it make sense to keep this section then?
 18. Figures 1, 4, and 5 and relevant supplementary figures: Define units for "Transformability" (it is not immediately obvious for a non-specialist). Define all error bars and the "ND" symbol ("not determined", "not detected", "not done"?).
 19. Figure 2a: Labels for insets b and c are swapped.
 20. Figure 2d,f: If I am not mistaken, these two panels are not referred to in the main text.
 21. Figure 2d: Orange on a cyan background (E. coli ProQ sequence) reads very badly. Consider choosing other colours of another means to highlight these residues.
 22. Figure 4d: Do the authors have an idea why the double mutant S70A-S72A does not bind RocR yet represses transformation quite well?
 23. L. 833: The description of the panel c of Supplementary Figure 1 does not correspond to the image: one does not see different loop sizes.
 24. Supplementary Figure 2: Calculating K_ds for RocR3nt and RocR7nt is pretty much meaningless: they are far from saturation, and the obtained values are misleading. How did the authors measure the K_d for RocR3bp, which has a double-band pattern? In general, it is striking that in many cases shown in panels d-g, at concentrations >2 µM the bound proportion does not change any more. Can it be that RocC starts to aggregate at such concentrations, which prevents reaching binding saturation? Or is it an RNA misfolding problem?
 25. Supplementary Figure 3: It looks as if the authors have swapped the labels for the curves. In panel a, the violet curve seems to correspond to RocC14-126 : RocRSL3, the green one is RocRSL3, and the blue one is RocC1-126. In panel b, the violet curve could be RocC1-126 : RocR9bp-tet, the green one is RocR9bp-tet, and the blue one is RocC14-126. Is that correct?

Alexandre Smirnov

We thank the reviewers for their thoughtful and constructive review of our manuscript. It is especially gratifying, given that we have been working for 20 years to get a structural view of one of these proteins bound to their RNA targets! Below are the reviewers' comments with our responses in blue.

Reviewer #1 (Remarks to the Author):

In this study, Kim et al. report the structure of the ProQ/FinO-domain protein RocC from *Legionella pneumophila* along with part of the bound small RNA RocR. The authors test predictions from their structure by assaying RNA binding of mutant proteins in vitro (via fluorescence polarization) and complementation in vivo (through the ability of mutant strains to take up DNA). The authors conclude the 3' nucleotide of the terminator is recognized by a conserved pocket in RocC, potentially explaining the differences in 3' tail recognition by other ProQ/FinO family proteins.

The data are complete and clearly presented and contribute to our understanding of this important family of RNA binding proteins. I only have relatively minor comments:

Page 10: Why did the authors not examine the consequences of mutating Gly52 given its conservation and proposed role in interacting with the 3' end?

The reviewer makes a good point here, however, the hydrogen bonding involves the peptide mainchain, and these interactions cannot be easily manipulated via mutation. Mutations at this position would likely perturb the β -turn structure, and therefore interactions with the RNA, however it would be difficult to tease out whether binding effects were really due to the loss of specific hydrogen bonding interactions, or through more extensive structural changes to the protein.

Figure 2b: I thought the double helix was very difficult to see in the figure and am hoping this can be improved by changing the colors/shading. There should be a key for the colors in these structure panels.

We have now added colored labels in the figure, and also added the hydrogen bonds between the base-pairs to hopefully clarify the double helix.

Figure 4d (and line 259): The authors need to be explicit about how they are expressing the mutant RocC derivatives in *L. pneumophila*. Are all of the mutant proteins present at the same levels?

We have included an Supplementary figure 10 showing western blotting for all the mutant proteins expressed in *L. pneumophila*. In most cases, the mutants are expressed at similar levels to WT, however we do find lower levels of expression for 3 of the point mutants that bind RNA least well: R75D, Y87F and R97M. Y87F is insoluble in vitro and so we think this mutation leads

to a folding defect, however the other mutants are highly soluble similar to WT. We suspect the lower levels of protein may be related to stabilization of both RocC and RocR upon RocC/RocR interaction. This is described in the revised manuscript: line 302-310.

Figure 6: While this figure provides a review of the function of ProQ/FinO-domain proteins, it does not summarize what is discovered in this study. I suggest the authors replace this figure and the description of it with more discussion of how the different ProQ/FinO-domain proteins bind tails of different lengths, possibly with a model illustrating the findings regarding the length specificity.

This is also a good point and it is something we struggled with. At the reviewer's suggestion, we have added a panel to figure 5 that illustrates our idea that ProQ/FinO domains might potentially recognize RNA targets with tails between 3 and 8 nucleotides in length. The 3 nucleotide minimum is defined by the hook structure that allows the 3' nucleotide to access its binding pocket, while the 8 nucleotide maximum is the number of nucleotides separating the 3' nucleotide from the nucleotide directly bound by the N-cap motif.

Reviewer #2 (Remarks to the Author):

The group of RNA binding proteins containing a ProQ/FinO domain have been recognised as an important class of RNA chaperones and mediators of regulatory RNA mediated control of gene expression in several bacterial species. How the proteins interact with RNA is not known precisely, and the work by Kim et al has now illuminated the atomistic details and elaborated rules for recognition. Using x-ray crystallography together with biophysical methods to test the observed contacts, the authors establish that a stem loop structure and 3' single stranded region with defined length and with 3'OH group are key determinants for the action of *Legionella pneumophila* RocC and its related homologs. The in vivo function is corroborated in an assay for transformability. The experimental work is well done and the paper is well written and the results explained clearly and insightfully. There are a few minor comments that will hopefully be helpful to for the authors to consider:

It may be helpful to explain whether the full length RocC protein contains other domains, even if these are intrinsically disordered, and whether these could potentially cooperate with the ProQ/FinO domain.

Great point. The C-terminal region of RocC, which is predicted to be intrinsically disordered, is required for repression of competence, but not for recognition of the terminator RNA. We think this region is critical for the mysterious "RNA remodeling" activity, and we think it is likely that other members of the family likely work in a similar manner. This is discussed in the final section of the Discussion (lines 405-434).

An analogous interaction of the helical N-cap interaction with phosphate occurs for phosphate

ion interaction at the active site of the exoribonuclease polynucleotide phosphorylase (as well as aracheal homolog). It might be helpful to comment on this.

Interesting. Indeed, I think the N-termini of helices often are used to interact specifically with phosphates – another common example are P-loop motifs. Our purpose here however was to really highlight the ability of this simple motif to recognize *consecutive* phosphates in a conformationally-specific manner. I think it is telling that we do not see recognition of consecutive phosphates in the protein-DNA database, even though recognition of single phosphate groups is very common within these proteins.

Line 176 The asymmetric unit contains 6 copies of apo-RoC14-126 and 4 copies of the protein-RNA complex. Are there any higher order arrangements with additional interactions of apo-RoC14-126 with the RocR9bp-tet RNA that could support higher order assembly?

Another interesting idea. We did look through the asymmetric unit and crystal packs for interesting additional interactions that might be biologically relevant. We do not really see anything that stands out. The C-terminus of $\alpha 4$ does seem to often pack into RNA minor grooves in the crystal pack. This can occur whether the protomer in question is bound to its own RNA or apo. The contact is quite small, involving ~ 3 H-bonds. We note that the complex does purify as a heterodimer, suggesting that additional interactions likely would be fairly weak. However our EMSAs with larger RNAs do suggest that additional protomers can load onto these RNAs at higher concentrations.

It might be that RocC will not bind RNA if the 3' end has a 2',3' cyclic phosphate. Is such a species likely to arise in any processing step of RNA targets?

I modeled a 2',3' cyclic phosphate on the terminal nucleotide and it does indeed clash with the β -hairpin motif. Such structures would be expected to arise as the product of certain ribonucleases (see Shigematsu et al. (2018) Front Genet.). This is now added to the Discussion (pg. 12).

In Figure 4, the Kds and transformability do not seem to scale in any simple way (compare for example S70A-S72A with K71D and K73D). Can the authors please comment on this?

The reviewer is referring to the surprising result that the S70A-S72A double mutant, which shows a dramatic loss in RNA binding *in vitro*, shows little if any effect *in vivo*. We really do not have a good explanation. The Ser residues only provide part of the recognition of the RNA. Lys71, Lys73, Arg75, as well as the N-cap mainchain NH groups also contact the contiguous phosphates along this section of the strand. Might it be that these interactions are sufficient for a biological response, even though the Ser mutations do significantly reduce the affinity *in vitro*? Part of the issue might be that the Ser mutations are simply removing a favorable interaction. The Lys and Arg mutations are charge swap mutations that remove a favorable interaction and replace it with an unfavorable repulsion. For that reason, one might expect the Lys and Arg mutations to have a larger effect than the Ser. We do note the discrepancy in the

Results section (pg 10) but we do not have an explanation for this.

Line 141 here after -> hereafter ?

Corrected

Figure 1a – here and elsewhere, is the inset for confidence for prediction necessary, since it is 1 throughout? Same for other figures too?

Good point. We removed this.

Figure 2, label secondary structural elements in panel a to match labels in panel d?

Done

Reviewer #3 (Remarks to the Author):

The manuscript “Structural basis for recognition of transcriptional terminator structures by ProQ/FinO domain RNA chaperones” by Kim et al. provides important structural, biochemical, and functional data about the way the Legionella ProQ homologue RocC recognises its target sRNA and represses the competence regulon in this bacterium. The ProQ/FinO family of RNA chaperones are widespread among gamma-, beta- and alpha-proteobacteria, where they play diverse roles ranging from narrow-range plasmid conjugation control to competence regulation to global post-transcriptional gene expression programming. While the biochemical basis and regulatory rationale of ProQ-like chaperones have been extensively studied, and their various modes of action are getting relatively well understood, some critical aspects of ProQ-RNA interactions remain elusive. In particular, it has not been known how ProQ-like chaperones can sense the structure of their RNA ligands and ‘measure’ the length of the 3’-terminal U-rich tail, which are two critical features of their RNA recognition mechanism distinguishing them from another major type of bacterial chaperones, Hfq. By combining X-ray crystallography, biochemical and genetic assays, the authors convincingly show that RocC, interacting nearly exclusively with the sugar-phosphate backbone on one side of the target RNA, can read its A-helical shape and specifically recognise, by enforcing an unusually distorted conformation of a short, 4-5-nucleotide-long single-stranded tail, its 3’-terminal OH-group. The authors demonstrate that this binding mode is critical for RocC function in vivo. Finally, they explore the effect of the 3’-terminal single-stranded tail on RNA binding by two other major representatives of the family, FinO and enterobacterial ProQ. These experiments bring about better understanding of how these proteins can select their targets among a wide variety of available ligands. The authors have generated an impressive amount of high-quality data, and their findings are extremely valuable to the field: this structure is truly a game-changer in our knowledge of how ProQ-like proteins engage their targets. I would only propose a couple of minor changes which, however, will certainly be required to bring the manuscript to a publishable level.

Many thanks for the extensive review and kind words.

1. The authors rapidly narrow down their attention to SL3, based on the affinity argument and for the sake of easier crystallisation. However, similar binding strength shown by RocR and SL3 alone does not necessarily mean that other parts of either molecule (RocR or RocC) do not participate in the interaction. A relevant example is Hfq-dependent sRNAs, which interact with Hfq primarily via the proximal face, and this binding is critical for their stability and largely defines affinity. However, nearly all sRNAs have additional binding regions engaging either the rim or the distal face of Hfq. These interactions do not contribute much to affinity but play a vital role in their functionality. I think the authors should be cautious about such possible multi-site interactions (e.g. involving the C-terminal domain of RocC and SL1 or SL2). While the binding mode they describe in this manuscript is certainly the most important one, it is likely not the only one in the context of the full-size molecules.

The reviewer makes a good point here – these proteins must contact RNA in a more complicated manner than what is shown in our minimal structure. I think it is clear that the ProQ/FinO domain does serve as a binder for transcriptional terminator structures however, in the cases where it has been studied, this domain is not sufficient for biological activity. Almost all these proteins have large intrinsically disordered regions that are essential. In the case of FinO, there is strong evidence that the flexible N-terminal region also contacts RNA and is essential for RNA duplexing and strand exchange. For ProQ, there is HDX that suggests that the C-terminal Tudor domain contacts RNA and our early work suggested this same region was necessary for RNA duplexing in vitro. RocC also has an extensive C-terminal flexible region that is also required for biological function. We discuss all this in the final section of the Discussion, and I think this point is also made in the final cartoon figure that gives an overview of how we think these proteins work. I have revised this section to clarify the point that additional regions likely contact RNA to facilitate RNA-RNA association (Pg. 15)

2. Looking at Figure 2b, I wonder whether having a longer helix 2 (continuing more toward the original N-terminus) could enable additional contacts between RocC and the base-paired portion of SL3. What do authors think about it? Can such an interaction, if any, be modelled?

We just do not have enough information at this point to model any additional interactions for the RocC/RocR complex. It is interesting to speculate about this, however. Our integrative SAXS modeling of the FinO-FinP SLII complex predicted the same overall protein-RNA orientation that has been revealed in the RocC-RocR crystal structure. In the FinO case, there is an interesting Trp at the tip of the N-terminal helix that we predicted could interact with the loop of the FinP hairpin. This Trp is important for in vivo function and for RNA duplex and RNA strand exchange activities of FinO. We suggested this might play a role in hairpin destabilization and/or stabilization of an RNA-RNA kissing complex. We used FRET with a fluorophore attached to the tip of the N-terminal helix and we were surprised to find that we did not get efficient FRET with the tip of the hairpin, but instead had efficient FRET with the base of the hairpin stem. This might suggest that the long helix in FinO folds down around the base of the stem but more will clearly be needed to really understand how larger protein and RNA constructs interact.

3. Ll. 32-33: More correct would be “bound to the transcriptional terminator of its primary partner, the sRNA RocR”.

Corrected

4. Ll. 67-68: sRNAs control not only stress responses but many other things too (e.g. competence), and this should be reflected in this sentence. Moreover, not all of them work by base-pairing with mRNA targets (although most certainly do).

Good points. The first paragraph has been edited to reflect this.

5. L. 79: “has enabled the elucidation”.

Done.

6. Ll. 80-83: The unusual situation described in these two sentences is in a way unique to Salmonella Typhimurium strain SL1344, which has three plasmids with a somewhat entangled relationship between their FinO-like regulators and their sRNA ligands. I think it is important to make it clear and probably better explain what is going on in this specific case (for the sake of clarity).

I have edited the introduction in an attempt to relate a little of this without straying too far from the major point.

7. L. 162: The reference to Figure 1c is not suitable here.

Right. Changed to Figure 1d.

8. L. 167: Should be “Figure 1d...”

Fixed

9. L. 172: The sentence is ambiguous. If I got it right, I would insert a comma here: “using gel filtration chromatography, and SEC-MALS confirmed...”

Thanks, corrected

10. Ll. 283-285, 376-378, Fig. 4d: I think the authors overstate the effect of the T82A mutation on transformability. It might be statistically significant, but it has certainly not a big size effect. This also agrees with the affinity measurement.

Agreed. Language has been changed in the indicated sections.

11. L. 314: Unless I am mistaken, the authors do not show results of this modelling. Do they refer to the published data cited just before in the same sentence?

We have done some modeling however we have decided that an analysis of this would be best left to a subsequent paper. I have changed the text to simply say that the structural and sequence similarities of the proteins mentioned suggest that they all likely bind RNA in similar ways. Interestingly, the most different might be NMB1681. It has some differences in the N-cap region that might indicate a different mode of interaction with the stem.

12. Section “Protein expression and purification”: I suppose the authors meant “100 µg/ml” and “35 µg/ml” for antibiotics concentration.

Indeed, that is what we meant! Fixed.

13. Same section, L. 510: It is not clear what “NaCl 3 0mM” exactly means.

30 mM NaCl. Fixed.

14. Ll. 595-596: Please, clarify/correct “Molecular Dynacmics ImageQuaNT TL software”.

Typo fixed.

15. For the two equations in Materials and Methods (ll. 598-599 and ll. 622-623), it would be useful to provide definitions (and units) for all symbols in more canonical, biochemical terms. For instance, the 4PL equation for FP data remains quite cryptic with respect to what one should measure or fit and how one calculates K_d values in the end.

We clarified these sections.

16. Ll. 618-619: “conducted in/on an Envision 2103 multilabel plate reader”?

Fixed

17. Section “Western blot analysis of RocC and RocC mutants”. Unless I am mistaken, western blotting data are not shown anywhere in the manuscript. Does it make sense to keep this section then?

This question is answered above.

18. Figures 1, 4, and 5 and relevant supplementary figures: Define units for “Transformability” (it is not immediately obvious for a non-specialist). Define all error bars and the “ND” symbol (“not determined”, “not detected”, “not done”?).

Done.

19. Figure 2a: Labels for insets b and c are swapped.

Corrected

20. Figure 2d,f: If I am not mistaken, these two panels are not referred to in the main text.

These panels are now referred to throughout the text.

21. Figure 2d: Orange on a cyan background (E. coli ProQ sequence) reads very badly. Consider choosing other colours of another means to highlight these residues.

This panel has been modified to improve visibility.

22. Figure 4d: Do the authors have an idea why the double mutant S70A-S72A does not bind RocR yet represses transformation quite well?

See response to reviewer 2.

23. L. 833: The description of the panel c of Supplementary Figure 1 does not correspond to the image: one does not see different loop sizes.

Fixed

24. Supplementary Figure 2: Calculating Kds for RocR3nt and RocR7nt is pretty much meaningless: they are far from saturation, and the obtained values are misleading. How did the authors measure the Kd for RocR3bp, which has a double-band pattern? In general, it is striking that in many cases shown in panels d-g, at concentrations $>2 \mu\text{M}$ the bound proportion does not change any more. Can it be that RocC starts to aggregate at such concentrations, which prevents reaching binding saturation? Or is it an RNA misfolding problem?

The reviewer makes some good points here. We removed the K_D estimates for RocR3nt and RocR7nt. For RocR3bp we think there is an alternative folding form here, maybe not surprising since the predicted hairpin stem would only have 3 bp. For this RNA, we estimated the binding by quantitating only the major free band and the major bound species. And the reviewer is correct that we do usually see residual unbound RNA even at the highest protein concentrations. RocC(14-126) is very soluble so we think protein solubility is not the problem. RNA misfolding might be the issue. These are pretty simple RNAs and you would expect three possible forms – hairpin, unfolded single strand, and dimer. All three should readily separate in our native gels (indeed we can make the dimer by snap-annealing the RNA). With the exception of RocR3bp, we only see one form of free RNA so we assume this is the hairpin. I really do not have a good explanation for what is going on. One possibility might be that this just is a feature of running EMSA on relatively weak, dynamic complexes. It does seem that this phenomenon is most pronounced for the weaker complexes (compare Supp Fig 2d with 2g). Perhaps complexes with a higher off-rate dissociate during the earliest phases of electrophoresis before the complexes enter the gel?

25. Supplementary Figure 3: It looks as if the authors have swapped the labels for the curves. In panel a, the violet curve seems to correspond to RocC14-126 : RocRSL3, the green one is RocRSL3, and the blue one is RocC1-126. In panel b, the violet curve could be RocC1-126 : RocR9bp-tet, the green one is RocR9bp-tet, and the blue one is RocC14-126. Is that correct? We have fixed this in the revised figures.

Alexandre Smirnov

REVIEWERS' COMMENTS

Reviewer #1 (Remarks to the Author):

The authors have made a commendable effort to address the reviewers' comments. I particularly like the panel added to Figure 5, though I suggest that the representation of the RNA be consistent between Figures 4 and 5. In my view this study is an important contribution to the field.

Reviewer #2 (Remarks to the Author):

The authors have provided detailed and compelling responses to the reviewer comments and have improved the manuscript.

Reviewer #3 (Remarks to the Author):

Most raised points have been properly addressed. I only spotted two requested changes that have not been introduced in the figures:

- 1) The Reviewer #2 requested, and I concur with this request, to label the secondary structure elements in Fig. 2a. This would enormously facilitate navigation through the structure.
- 2) I suggested removing Kd estimates for RocR3nt and RocR7nt from Supplementary Fig. 2c as unreliable. But they are still there.

Excellent revision otherwise!

Alexandre Smirnov

We thank the reviewers for constructive comments. Below are the reviewers' comments with our responses in blue.

Reviewer #1 (Remarks to the Author):

The authors have made a commendable effort to address the reviewers' comments.

I particularly like the panel added to Figure 5, though I suggest that the representation of the RNA be consistent between Figures 4 and 5.

Good point. We corrected the RNA to be consistent between Figures 4 and 5.

In my view this study is an important contribution to the field.

Reviewer #2 (Remarks to the Author):

The authors have provided detailed and compelling responses to the reviewer comments and have improved the manuscript.

Reviewer #3 (Remarks to the Author):

Most raised points have been properly addressed. I only spotted two requested changes that have not been introduced in the figures:

1) The Reviewer #2 requested, and I concur with this request, to label the secondary structure elements in Fig. 2a. This would enormously facilitate navigation through the structure.

Agree, corrected.

2) I suggested removing Kd estimates for RocR3nt and RocR7nt from Supplementary Fig. 2c as unreliable. But they are still there.

Thanks, corrected.

Excellent revision otherwise!

Alexandre Smirnov